# Long-Term Behavior of Concrete Containing Wood Biomass Fly Ash

**Ivan Gabrijel ***[ID]**, Marijan Skazlić and Nina Štirmer** [ID]

Faculty of Civil Engineering, University of Zagreb, Fra Andrije Kačića Miošića 26, 10000 Zagreb, Croatia
*  Correspondence: ivan.gabrijel@grad.unizg.hr; Tel.: +385-1-4639-440

**Abstract:** Wood biomass is widely used in the European Union as a fuel for the production of heat and electrical energy, generating a considerable amount of ash. The disposal of ash, especially its finest fraction, requires proper engineering solutions, since these particles contain heavy metals and can easily pollute soil, groundwater, or air. In this work, wood fly ash with a high amount of pozzolanic oxides and one with a high CaO content were used in concrete as a 15% and 30% cement replacement. Incorporation of wood ash in concrete reduced the 28-day compressive strength of concrete by up to 37%, which was attributed to the low stiffness of the wood ash particles, while the 2-year compressive strength indicated very low pozzolanic reactivity. The capillary absorption of concrete increased with the increase in the ash content, but almost no influence on the gas permeability was observed. Wood fly ash with high CaO content reduced the drying shrinkage of concrete by up to 65% after 1 year. In a mix with 30% of high CaO fly ash, swelling occurred in the first days of hydration, which was attributed to the volume expansion due to the formation of portlandite and brucite, but did not lead to cracking or a decrease in long-term compressive strength.

**Keywords:** capillary absorption; compressive strength; drying shrinkage; supplementary cementitious materials; wood biomass fly ash





## 1. Introduction

In the European Union (EU), the tendency to reduce greenhouse gas emissions from coal use has led to increased use of wood biomass for energy production [1]. It was estimated that in 2018, about 11 million tons of ash were generated from biomass combustion in the EU-28 countries, most of which was ash from wood biomass [2]. It is very important that ash from wood biomass is well managed to avoid the possible pollution of air, soil, or groundwater. However, most of the ash produced is currently landfilled due to logistical problems, different ash properties, or the lack of legislation [3–6]. Ash from the natural wood combustion contains valuable plant nutrients that are concentrated in the coarse ash fractions, so perhaps the best way to manage this ash is to return it to the forest from which it originated [7]. The fine ash fractions, fly ash, contain most of the volatile heavy metals and should be utilized in industry or disposed of [7].

It has been shown that fly ash can be used as a raw material for the production of various types of construction products [8–13]. Today, fly ash from coal combustion is one of the main cement replacement materials in cement production, and its use as a concrete constituent has been standardized [14]. In addition to coal fly ash, many new supplementary cementitious materials are being investigated to increase the sustainability of cement-based products either by increasing the durability or by reducing the cement consumption [15–18]. The cement and concrete industry has also been identified as one of the main potentials for the use of wood fly ash (WFA) [6]. The properties of WFA differ from those of coal fly ash. The differences in the chemical composition of WFA are related to the wood species, the wood pieces used as fuel, and the season of biomass harvest [19,20]. In addition, different combustion technologies have a great influence on

the physical and chemical properties of WFA particles. Although they are not as reactive as cement, WFA from grate combustors and fluidized bed combustors can exhibit hydraulic and/or pozzolanic properties [21–23]. Extensive research has been conducted to investigate the effect of using WFA as a cement replacement material [20–27]. When used as a cement replacement material, WFA deteriorates the mechanical properties and increases the water demand, which is attributed to the amount of unburned carbon particles that have lower stiffness compared to cement particles. However, replacing cement with WFA can also increase the strength and improve the workability due to improved particle packing or activation by grinding [2,23,28].

The utilization of WFA in the concrete industry requires further research, especially since most studies on the influence of WFA on the properties of cement composites have been tested on cement pastes and mortars, and only a limited number of experiments have been scaled up to the concrete level [2,26,29]. In this work, two types of WFA with different physical and chemical properties are used as partial cement replacements in concrete. The effects of the WFA admixture on the compressive strength, permeability, and drying shrinkage of concrete measured over a period of one year were studied.

## 2. Materials and Methods

### 2.1. Properties of Wood Fly Ash

Wood fly ash (WFA) was collected from two power plants in Croatia. Both plants are cogeneration biomass power plants that produce heat and electrical energy and use the same technology of incineration by grate combustor. Power plant 1, which produces WFA1, is located in the northern part of Croatia and has a production capacity of 2.75 MW of electrical energy and 15 MW of heat energy. The types of wood used as fuel in Plant 1 are beech, oak, fir, and spruce, which are used in the form of wood chips from roundwood and thinning residues including twigs and tops, branches, bark, needles/leaves, and wood industry waste (including bark). Plant 2, which produces WFA2, has an electric capacity of 1 MW and a heat capacity of 4.1 MW, and is located in the mountainous part of Lika−Senj County. Plant 2 uses beech, oak, and hornbeam as fuel in the form of wood chips made from roundwood and thinning residues including twigs, tops, and branches. Fly ash particles from the combustor are collected by bag filters [30]. The data on the incineration temperature of wood biomass came from the power plant technologists and range from 700 to 950 °C in Plant 1 and up to 800 °C in Plant 2.

The chemical properties of the WFAs used in this work are presented in Table 1. Elemental composition was determined by X-ray fluorescence according to the standard ISO/TS 16996:2015, loss on ignition (LOI) according to ASTM D7348-13, and pH value according to EN 12176:2005 [31–33]. WFA1 has a CaO–Al$_2$O$_3$–SiO$_2$ ratio in the range of coal fly ash [2,34] and a pozzolanic oxide content higher than the minimum value for class C pozzolans according to ASTM C618-19 [35]. WFA2 had a high CaO content and a relatively low amount of pozzolanic oxides. The density was determined according to ASTM C188-17 [36]. WFA1 contained a significant amount of unburned wood particles, which made accurate density determination difficult because these particles tended to float on the petroleum. WFA2 did not contain any unburned wood particles.

Particle-size distribution (PSD) was analyzed by a laser diffraction method using a dry measurement (Shimadzu SALD 3101 instrument, Kyoto, Japan). The median particle size ($d_{50}$) of WFA1 ranged from 107 to 144 µm and of WFA2 from 57 to 72 µm. Cement had finer particles than WFA with $d_{50}$ ranging from 9.0 to 9.6 µm.



**Table 1.** Chemical composition and density of the cement and WFA.

| Property | CEM | WFA1 | WFA2 |
|---|---|---|---|
| $P_2O_5$ | 0.22 | 1.40 | 1.97 |
| $Na_2O$ | 0.85 | 2.12 | 0.57 |
| $K_2O$ | 1.25 | 5.28 | 8.10 |
| CaO | 59.80 | 18.58 | 57.93 |
| MgO | 2.01 | 3.68 | 6.17 |
| $Al_2O_3$ | 4.94 | 12.42 | 3.14 |
| $TiO_2$ | 0.23 | 1.21 | 0.13 |
| $Fe_2O_3$ | 3.15 | 4.78 | 2.1 |
| $SiO_2$ | 21.88 | 49.34 | 18.19 |
| $SO_3$ | 3.33 | 1.17 | 1.70 |
| Pozzolanic oxides ($SiO_2 + Al_2O_3 + Fe_2O_3$) | 29.97 | 66.54 | 23.43 |
| Alkalis ($Na_2O + 0.658\,K_2O$) | 1.67 | 5.59 | 5.90 |
| LOI (at 950 °C) | 3.6 | 6.0 | 3.0 |
| pH | 12.9 | 12.9 | 13.2 |
| Density | 3.1 | 2.0 | 2.7 |

### 2.2. Concrete Mix Design

WFA was stored in the laboratory for a period of about two months before being used in concrete. During storage, WFA was sealed in plastic bags and then stored in closed plastic containers.

The cement used in this study was Portland cement CEM I 42,5 R, according to the European standard EN 197-1:2011 [37]. The aggregate was crushed dolomite with an average bulk density of 2.8 kg/dm$^3$ and an absorption of 0.5%, 0.7%, and 0.3% for fractions 0/4, 4/8, and 8/16 mm, respectively. The aggregate grading curves were the same as presented in [2] and are therefore not repeated here.

The composition of the concrete mixes is given in Table 2. The mixes had the same water/(cement + WFA) ratio of 0.5. Mixes containing WFA were prepared with cement replacement percentages of 15% and 30%. Mixes M1 and M2 were made with WFA1 and mixes M3 and M4 with WFA2.

**Table 2.** Composition of the concrete.

| Mix Designation | M0 | M1 | M2 | M3 | M4 |
|---|---|---|---|---|---|
| Cement (kg) | 400 | 340 | 280 | 340 | 280 |
| WFA cement replacement (%) | 0 | 15 | 30 | 15 | 30 |
| WFA content (kg) | 0 | 60 | 120 | 60 | 120 |
| Cement + WFA (kg) | | | 400 | | |
| w/(cem. + WFA) ratio | | | 0.5 | | |
| Water (kg) | | | 200 | | |
| Aggregate (kg) | 1831 | 1802 | 1780 | 1805 | 1796 |
| Fine aggregate (kg) | 652 | 641 | 634 | 642 | 639 |
| Coarse aggregate (kg) | 1179 | 1161 | 1146 | 1163 | 1157 |

All constituents were conditioned to a temperature of 20 ± 2 °C before mixing. Mixing was carried out in a compulsory mixer with a capacity of 75 L, following the mixing procedure specified in the standard EN 480-1:2014 for the preparation of reference concrete [38]. First, the aggregates and about half of the water were mixed for 2 min. After a two-minute break, mixing was continued with the addition of cement and WFA. After 30 s, the remaining water was added and mixing continued for 2.5 min.

### 2.3. Testing Methods

The following concrete properties were measured in the fresh state: consistency by slump test (EN 12350-2:2019), density (EN 12350-6:2019), air content (EN 12350-7:2019), and temperature [39–41]. The concrete was compacted on a vibrating table. After compaction,

the specimens were stored in a room with a temperature of 22 ± 2 °C and covered with a plastic sheet. After 24 h, the specimens were demolded and moved to a curing room where they were cured in air at a temperature of 20 ± 2 °C and a relative humidity of >95%. All specimens for the fresh and hardened concrete tests were from the same batch.

Concrete compressive strength was measured on concrete cubes with a side length of 150 mm at 28 days of age using a standardized method (EN 12390-3:2019) [42]. In addition, compressive strength was tested on specimens cut from prisms used to measure the drying shrinkage. The main reason for this additional test was the increase in length found in the specimens from the M4 mix. From each prism, three concrete cubes with a side length of 10 cm were sawn (Figure 1a). Thus, a total of nine specimens from each mix were tested. At the time of testing, the specimens were 2 years old. The compressive strength measured on cubes with a side length of 10 cm was converted to the strength of a 15 cm cube by multiplying by the coefficient 0.95.

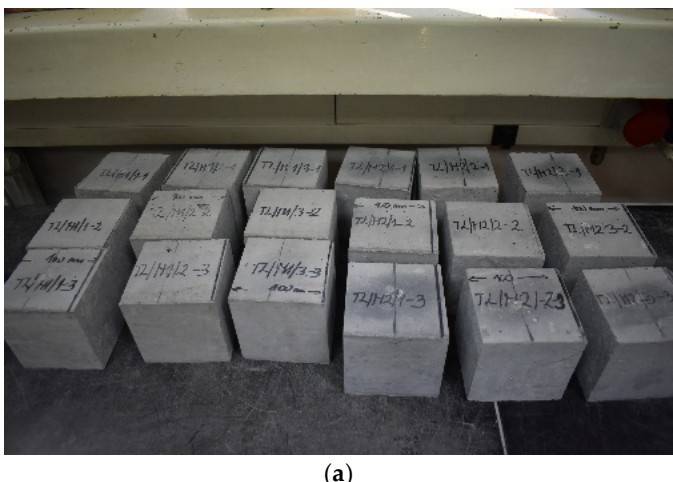 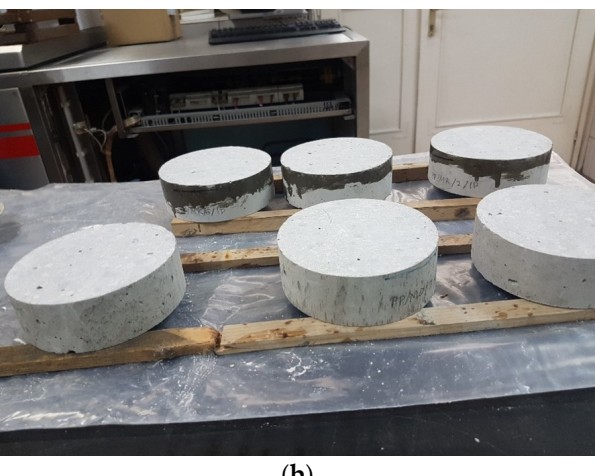

(**a**)                                                                    (**b**)

**Figure 1.** Preparation of samples for testing. (**a**) Cubes sawn from concrete prisms from the M1 and M2 mixes. (**b**) Coating of samples for capillary absorption measurement with epoxy resin.

The measurements of capillary absorption and gas permeability were performed on cylindrical specimens with a diameter of 150 mm and a height of 50 mm, obtained by sawing from a standard cylinder with a diameter of 150 mm and a height of 300 mm. Before sawing, the concrete cylinders were cured in a curing room for 28 days. Upper and bottom slices of the cylinder were not tested to avoid the effects of different boundary conditions. For capillary absorption measurements, the side surface in contact with water was coated with epoxy resin (Figure 1b). Before testing, the specimens were oven-dried at a temperature of 105 ± 5 °C until the change in mass was less than 0.5 g (≈0.025% of mass) for two consecutive weighings, and then allowed to cool to ambient temperature in a sealed container. Specimens for capillary absorption measurement were placed on cylindrical rods in a water container, and the water level was adjusted so that the bottom of the specimens was immersed 2–5 mm into the water. The mass of the specimens was measured at intervals of 5, 15, 30, 60, 120, 240, and 1500 min. Gas permeability was measured using a permeability testing device (SO2000H, Testing Bluhm & Feuerberdt GmbH, Germany). Nitrogen was used as the permeating fluid at a pressure of 2.0, 2.5 and 3.0 bar. Both gas permeability and capillary absorption were determined on three specimens from each mix.

The porosity and densities of the concrete were determined by the vacuum saturation method [43,44]. Before vacuum saturation, specimens were oven dried and cooled to room temperature in a closed container. For each mix, five specimens (cylinders 150 mm in diameter and 50 mm in height) were placed in a desiccator and exposed to a vacuum of 760 mm Hg for 3 h. After 3 h, the deaerated water was drained into the desiccator, and the vacuum was maintained for another hour. The specimens were then removed from the desiccator and kept under water until the mass of the specimens had changed by less

than 0.5 g after two consecutive measurements. The specimens were then weighed in the surface-dry condition in air and in water. The bulk density in the dry ($\rho_{z,dry}$) and saturated conditions ($\rho_{z,sat}$), apparent solid density ($\rho_a$), and apparent porosity ($p_a$) were calculated according to Equations (1)–(4). Since it was assumed that only open pores were filled with water, the term apparent is used here.

$$\rho_{z,dry} = \frac{m_{dry}\rho_w}{m_{sat} - m_{sat,w}} \tag{1}$$

$$\rho_{z,sat} = \frac{m_{sat}\rho_w}{m_{sat} - m_{sat,w}} \tag{2}$$

$$\rho_a = \frac{m_{dry}\rho_w}{m_{dry} - m_{sat,w}} \tag{3}$$

$$p_a = \frac{m_{sat} - m_{dry}}{m_{sat} - m_{sat,w}} \cdot 100 \tag{4}$$

In Equations (1)–(4), $m_{dry}$ is the mass of dry material, $m_{sat}$ is the mass of saturated material, $m_{sat,w}$ is the mass of saturated material weighed in water, and $\rho_w$ is the density of water (1000 kg/m$^3$).

Drying shrinkage was measured on prisms with dimensions $100 \times 100 \times 400$ mm. After demolding at an age of 24 h, measuring pins were glued at the distance of 200 mm on opposite sides of each prism and an initial measurement was made. Specimens were stored in a chamber at a temperature of $20 \pm 2$ °C and relative humidity of $65 \pm 5\%$. The change in length was measured with a length comparator, and the displacement was read with a Marcator digital indicator type 1086 (Mahr GmbH, Göttingen, Germany) (Figure 2).

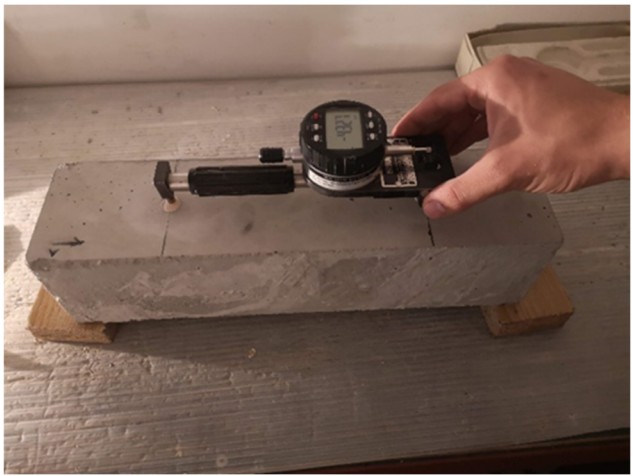

**Figure 2.** Measurement of the length change on the concrete prism.

For the TG analysis, concrete powder was extracted from the samples of 2-year-old concrete. To avoid a possible carbonated surface, the powder was extracted with a Profile grinder apparatus (Germann Instruments, Copenhagen, Denmark) from a depth of 10 to 17 mm. Thermogravimetric analysis was performed on the samples of $50 \pm 5$ mg heated from 30 °C to 1000 °C at a rate of 10 °C/min and a nitrogen flow of 40 mL/min using a TGA 55 instrument (TGA 55, TA Instruments, New Castle, DE, USA).

## 3. Results and Discussion

### 3.1. Fresh Concrete Properties

The density of the fresh concrete was determined using a measuring container with a capacity of 8 L. The density varied between the mixes in the range of 2440–2460 kg/m$^3$

(Table 3). The air content was low in all mixtures, indicating dense packing of the concrete constituents.

**Table 3.** Properties of the fresh and hardened concrete.

| Mix Designation | M0 | M1 | M2 | M3 | M4 |
|---|---|---|---|---|---|
| Fresh density (kg/m$^3$) | 2450 | 2460 | 2440 | 2460 | 2450 |
| Air content (%) | 1.5 | 1.4 | 1.5 | 1.0 | 1.1 |
| Slump (mm) | 80 | 60 | 50 | 60 | 20 |
| Measured initial temperature (°C) | 23.5 | 26.8 | 25.0 | 25.1 | 25.0 |
| Calculated initial temperature (°C) | 19.7 | 21.9 | 20.8 | 19.7 | 18.1 |
| ΔT (°C) | 3.9 | 4.9 | 4.2 | 5.4 | 6.9 |

The initial temperature of the concrete was between 23.5 and 26.8 °C. The temperature of all the constituents was measured before mixing, and the average temperature of the mix was calculated taking into account the mass and specific heat capacity of the constituents. The calculated initial temperature is given in Table 3. For the calculation of the initial temperature of concrete, it was assumed that the specific heat capacity of cement, aggregate, and water was 750 J/(kg K) [45,46], 1000 J/(kg K) [47], and 4200 J/(kg K), respectively. It was also assumed that the specific heat capacity of WFA was the same as that of cement [45]. The difference between the calculated and the measured initial temperature of the concrete corresponded to the temperature rise during mixing (ΔT). All mixes containing WFA had a higher rise in temperature during mixing compared to the reference mix. The highest ΔT was observed in mixes containing WFA2. This could be partly due to the reduced workability, as determined by the slump test, which resulted in increased interparticle friction during mixing, and consequently an increase in the temperature of the mix. Another contribution to the temperature rise during mixing could be related to the content of alkalis, especially $K_2O$, in WFA (Table 1). Alkalis may increase the reactivity of the $C_3A$ present in the cement and increase the water demand, but would also contribute to early heat generation, leading to an increased temperature of the fresh concrete [22,48,49]. The formation of ettringite from $C_3A$ and its precipitation on the surface of cement particles or in the pore solution is the main cause of the gradual loss of workability in cement pastes [50,51]. Therefore, the high content of alkalis, especially in WFA2, could be the main cause of the increased water demand, as determined by the slump test (Table 3). Another contribution to the heat release in mixes with WFA2 could be the hydration of CaO, which has a heat of hardening of 1168 J/g and reacts quickly on initial contact with water [52]. Unburned carbon, which is the main content of the LOI in fly ash, is characterized by high porosity and large capacity for water absorption [53]. This also contributed to the slump decrease, especially in mixes with WFA1 (Table 3).

### 3.2. Density, Porosity, and Permeability

The apparent solid density shows that replacing cement with WFA results in a reduction in the mass of the solid, which is consistent with the lower densities of WFA compared to the cement and aggregates (Table 4). In general, the bulk density decreases with increasing cement replacement, but this parameter also depends on the amount of air entrapped during the compaction process. Porosity increased with the increase in fly ash content, although no general trend between the WFA content and porosity could be observed (Table 4). This is consistent with the results presented in [2], where a larger increase in apparent porosity was observed only for mixes with 45% cement replacement by WFA.

**Table 4.** Densities and porosity of concrete.

| Mix Designation | M0 | M1 | M2 | M3 | M4 |
|---|---|---|---|---|---|
| Bulk dry density (kg/m$^3$) | 2355.9 ($\pm$22.6) | 2349.7 ($\pm$10.0) | 2297.3 ($\pm$23.9) | 2362.1 ($\pm$17.4) | 2339.8 ($\pm$8.7) |
| Bulk saturated density (kg/m$^3$) | 2504.2 ($\pm$16.0) | 2493.9 ($\pm$6.3) | 2452.6 ($\pm$16.9) | 2503.5 ($\pm$14.2) | 2487.0 ($\pm$6.3) |
| Apparent solid density (kg/m$^3$) | 2766.0 ($\pm$5.7) | 2745.5 ($\pm$2.3) | 2719.9 ($\pm$8.7) | 2751.0 ($\pm$11.3) | 2743.7 ($\pm$11.7) |
| Apparent porosity (%) | 14.83 ($\pm$0.67) | 14.42 ($\pm$0.39) | 15.54 ($\pm$0.74) | 14.14 ($\pm$0.39) | 14.72 ($\pm$0.51) |

The numbers in brackets are standard deviations.

Average values of the gas permeability coefficients with the corresponding coefficient of variation calculated as the ratio of standard deviation and average value are plotted in Figure 3. Since all data in Figure 3 were within a range between $1 \times 10^{-16}$ and $2 \times 10^{-16}$ m$^2$ and taking into account that the coefficient of gas permeability for concrete spans over a several orders of magnitude from $<10^{-18}$ m$^2$ for very low permeable concrete to $>10^{-15}$ m$^2$ for very permeable concrete shows that the replacement of cement with up to 30% with WFA had a very small impact on the gas permeability [54,55]. For mix M2, the gas permeability coefficient slightly increased, which could be due to the loosening of the particle packing by the coarser WFA1 particles. However, this increase was also within the reproducibility limits for the gas permeability method, which expressed through the coefficient of variation was around 30% [56–58].

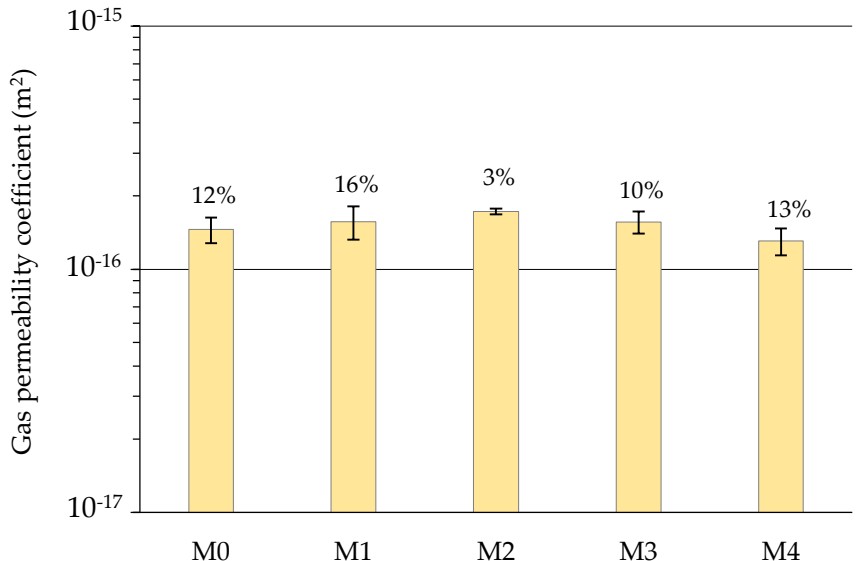

**Figure 3.** Results of the gas permeability measurements.

Results of the capillary absorption measurement are given in Figure 4. In mixes M2 and M3, a greater dispersion of results occurred, but regardless, the trend of an increasing capillary absorption rate with an increasing quantity of WFA was clear. Mixes with 30% cement replacement showed the highest capillary absorption rate. This may be attributed to the ability of WFA particles to absorb water, besides unburned carbon particles, since both fly ashes contain much coarser particles than cement (more than 50% of the particles were larger than 0.063 mm), which are probably porous and can absorb a certain amount of water. This could also be the reason for the lower slump of the fresh concrete.

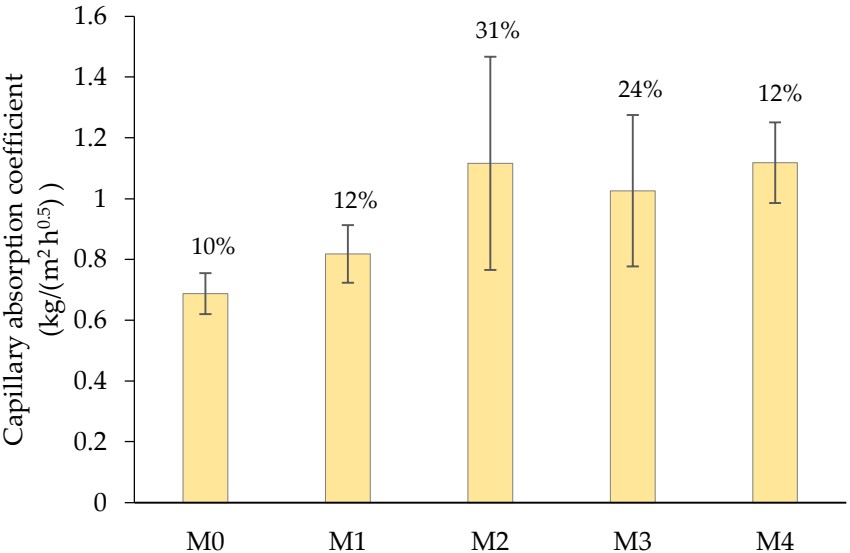

**Figure 4.** Results of the capillary absorption measurements.

### 3.3. Compressive Strength

Compressive strength was determined on three specimens for each mix. Figure 5 shows the average compressive strength measured after 28 days and the standard deviation. Replacement of cement with WFA resulted in a decrease in the compressive strength. Compressive strength also decreased with the increase in cement replacement. WFA2 contributed more to the decrease in strength compared to WFA1. It is interesting to note that mixes with WFA2 had a higher density and lower porosity than mixes with WFA1 (Table 4). WFA1 had coarser particles than WFA2 and cement, which loosened the particle packing. Although WFA2 allowed for the production of mixes with denser particle packing, it decreased the compressive strength. This could be explained by the lower stiffness of WFA2 particles compared to WFA1 [59].

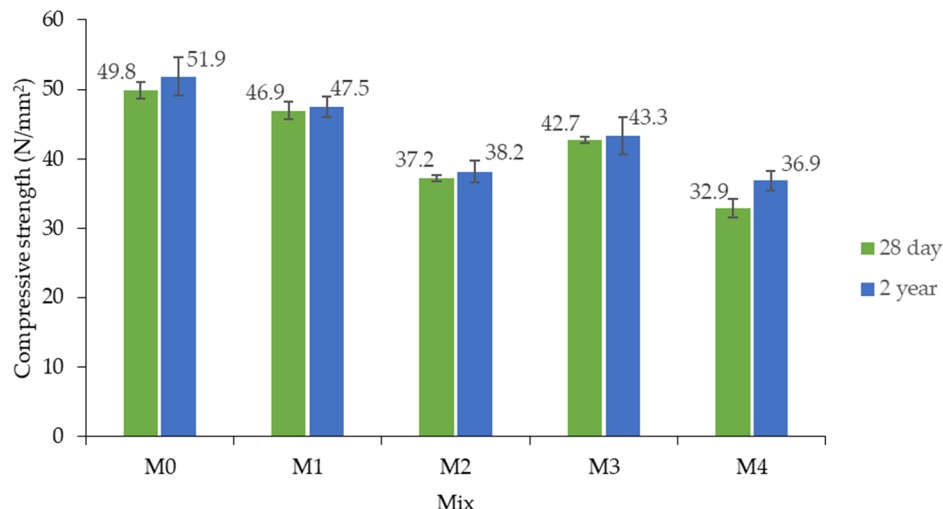

**Figure 5.** Compressive strength of concrete after 28 days and after 2 years.

A decrease in strength due to the replacement of cement with WFA has been reported by various researchers [20,28,49,60–62] and has been attributed to the lower content of cementitious material or the incorporation of weak, unburned carbon particles into the cement matrix. WFA are less reactive than cement, especially in short periods. It was discussed that within the first 28 days, most of the influence of cement replacement by WFA can be explained through filler and filling effects [2].

The compressive strength measured at 2 years of age corresponded very well to the 28-day strength. All mixes showed a slight increase in compressive strength compared to the 28-day strength. This indicates a very low reactivity of WFA used in this work. However, it should be noted that the curing conditions of the specimens were different. The standard concrete cubes were cured in air at a relative humidity >95%, while the specimens used for the drying shrinkage measurement were cured at a relative humidity of $65 \pm 5\%$.

### 3.4. Drying Shrinkage

The change in length of the concrete samples was monitored over a period of more than one year (395 days for mix M0 to 380 days for mix M4), with the first measurement taken at 24 h of age. Three samples of each mix were tested. Subsequently, the average value and the corresponding standard deviation of the change in length were calculated and are presented in Figure 6. Mix M1 with 15% cement replacement by WFA1 exhibited the same shrinkage deformation as mix M0 during the measurement period. For mix M2, the average shrinkage deformation decreased by about 37% after one year compared to the reference mix M0. It was found that increasing the aggregate/cement ratio of the concrete reduced the volume of the contracting cement paste and the overall shrinkage capacity [48]. Replacing cement with WFA could lead to a similar effect, since WFA is not as reactive as cement, as shown by the compressive strength test (Figure 5), but also by previous studies with wood ash from the same source [2]. Measurements on specimens from mixes M3 and M4 showed that WFA2 had a larger influence on the drying shrinkage behavior than mixes with WFA1. For mixes M3 and M4, where 15% and 30% cement were replaced by WFA2, shrinkage deformations after one year were reduced by about 32% and 65%, respectively, compared to mix M0. Mix M4 also showed an increase in the specimen length in the early aging phase. Since swelling of the specimens can be accompanied by cracking and weakening of the structure, additional investigations were carried out on the specimens used for drying shrinkage. Visual inspection of the specimens was performed periodically up to an age of the specimens of 2 years. Visual inspection did not reveal any cracks that would indicate deterioration of the specimens. Additionally, to verify that the initial swelling had not caused internal damage of the concrete structure, the compressive strength was measured as described in Section 2.3.

All mixes showed very similar trends in drying shrinkage at later ages, suggesting that the main difference between mixtures was due to early age reactions. Carevic et al. [27] analyzed the chemical properties of WFA from the same power plants as the WFA used in this work and found that WFA2 had a significant amount of free MgO and free CaO. Both free CaO and free MgO are recognized as the main causes of unsoundness of Portland cement [48]. However, they are also often used as expansive agents to reduce shrinkage deformations in cement-based materials [63–69].

In contact with water, free CaO forms $Ca(OH)_2$ (portlandite), followed by volume expansion. The reactivity of CaO depends on the calcination temperature. A higher calcination temperature results in less reactive CaO, but regardless of the calcination temperature, CaO reacts within the first few days if sufficient water is available for the reaction [70–72]. The increase in the length of the concrete prisms from mix M4 could have been caused by the formation of portlandite.

In Portland cement, the expansion of MgO is attributed to the formation of $Mg(OH)_2$ (brucite), which is formed by the reactions between magnesium and water [48]. This reaction proceeds very slowly and can lead to long-term unsoundness. MgO was found to be much more reactive at calcination temperatures up to 900 °C during early hydration than MgO in cement clinker calcined at ~1450 °C [73]. The WFA used in this work was burned at temperatures up to 950 °C. It is therefore possible that the MgO contained in the WFA was much more reactive than the MgO contained in Portland cement and could have contributed to the early age expansion of the concrete prisms.

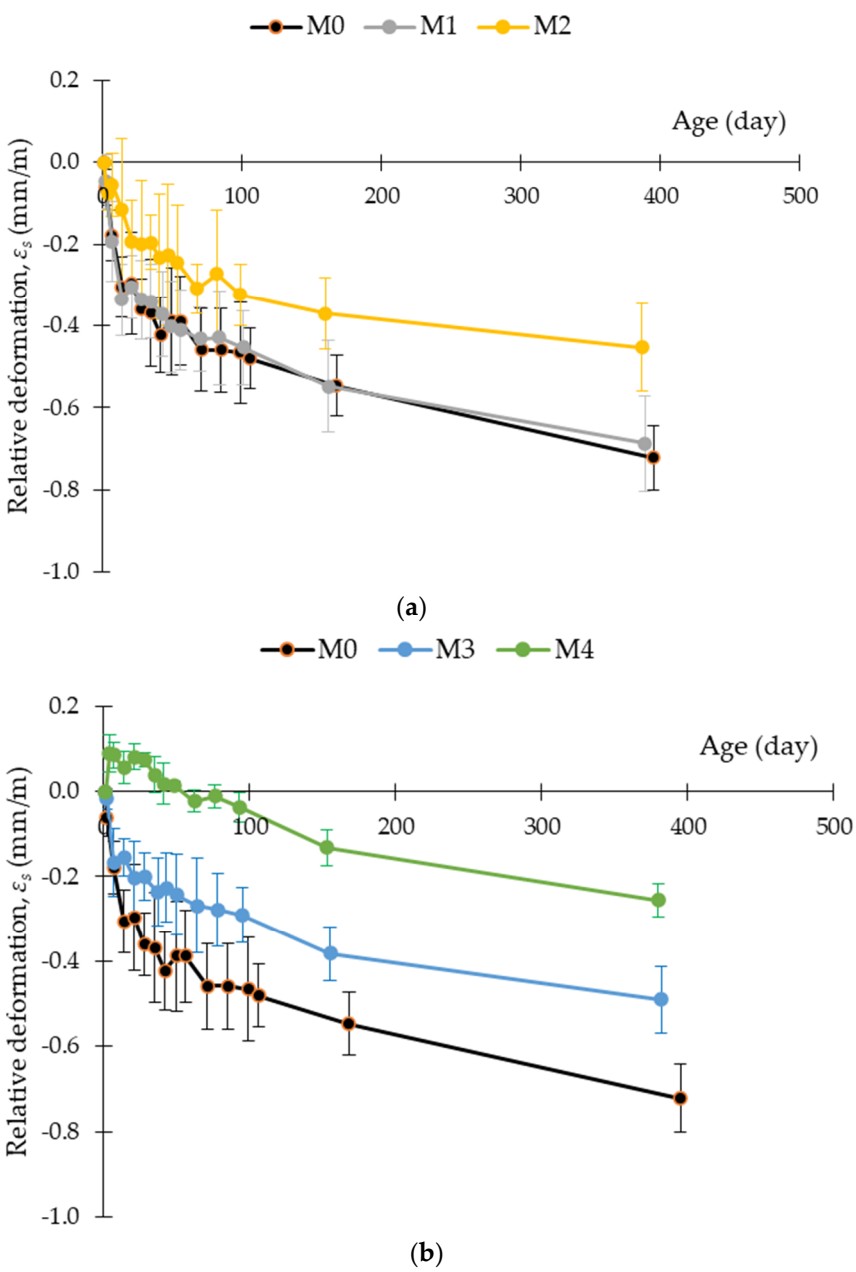

**Figure 6.** Results of the length change measurements: (**a**) Mixes M0, M1, and M2; (**b**) Mixes M0, M3, and M4.

Evidence for the presence of brucite in the concrete mixes studied in this work was additionally provided by thermogravimetric (TG) analysis. The DTG curves are shown in Figure 7. Two distinct peaks can be seen in Figure 7a. A smaller peak in the range between 400 and 450 °C indicates the decomposition of portlandite, while the large peak between 600 and 800 °C belongs to the decomposition of dolomite [74,75]. Since brucite decomposes in the temperature range of 350–450 °C [76], this part of the DTG curves is shown enlarged in Figure 7b. The peak occurring around 350 °C could indicate the possible decomposition of brucite. This peak is most pronounced in mixtures M3 and M4, which contain WFA2. This supports the assumption that brucite could partly be the cause of the initial volume increase.

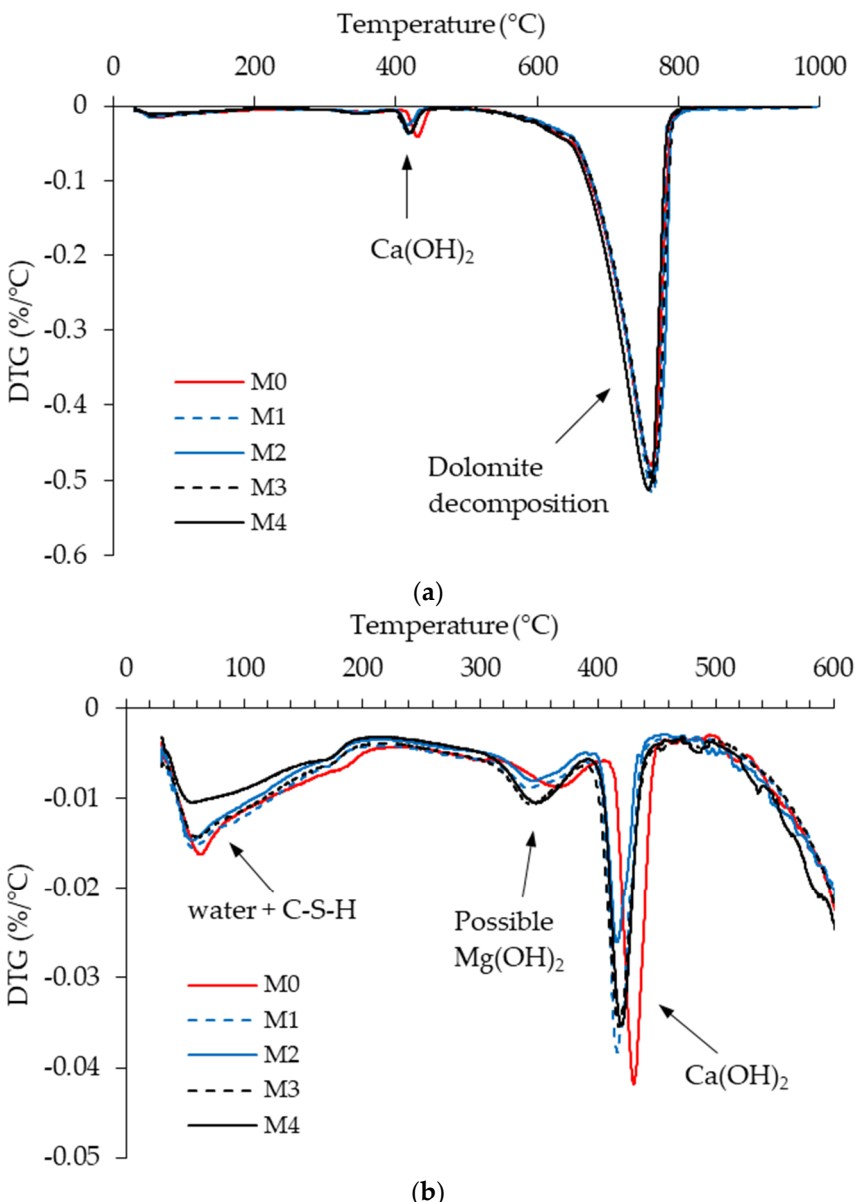

**Figure 7.** Results of the TG analysis. (**a**) DTG curves in the entire temperature range from 30 to 1000 °C; (**b**) enlarged DTG curves in the temperature range up to 600 °C.

However, free MgO from WFA is not the only possible source of brucite in the mixes analyzed in this work. In concrete made with dolomite aggregate, brucite ($Mg(OH)_2$) and calcite ($CaCO_3$) form as a result of the dedolomitization reactions of the reaction between dolomite and portlandite [77]. The rate of this reaction is influenced by the temperature and the content of alkalis [77,78]. The rate and extent of the reaction also depend on the particle size of the dolomite. Larger grains are almost always less reactive because they can form a protective shell of reaction products [78,79]. The number of particles passing the 0.063 mm sieve of the crushed dolomite aggregate used in this work was about 2.5% of the concrete mass. It can therefore be assumed that brucite was present in all the mixes analyzed in this work. It has been discussed that the dedolomitization reaction does not cause expansion [80,81]. For the aggregate used in this work, there have been no previous reports that alkali aggregate reactions cause any volume instabilities, which was also further confirmed by the drying shrinkage results. Therefore, the cause of the volume increase is probably due to the reactions caused by the wood ash.

Figure 8 shows an example of detrimental cracking in concrete caused by the expansive effect of wood ash in concrete (results of a previous study [82]). The sample in Figure 8 was made with a 30% replacement of Portland cement with WFA containing 57.6% CaO and 0.7% MgO. In the same study, no cracks were found in the concrete mix made with 15% of cement replacement [82]. The probable mechanism that leads to the cracking of the sample in Figure 8 is related to the crystallization pressures caused by expansion during the hydration of free CaO [83]. The formation of $Ca(OH)_2$ from CaO is accompanied by the molar volume increase by a factor of $\approx 2$ [84]. The growth of $Ca(OH)_2$ crystals in the pore space fills the voids without causing much expansion of the cement paste, but as crystals become enclosed, their expansion becomes restrained and stresses build up. The energy forcing the $Ca(OH)_2$ crystals to grow can lead to pressures exceeding 150 MPa in fully restrained samples [85]. In the cement paste matrix, crystallization pressures are controlled by the amount and reactivity of free CaO, the possibility of the diffusion of $Ca^{2+}$ and $OH^-$ ions through the pore space, and the degree of restraint [86]. During storage, the amount of free CaO and free MgO in WFA decreases due to reactions with $CO_2$ or water present in the air [27]. In the case of concrete from Figure 8, WFA was used in the concrete two weeks after sampling at the power plant, and the first cracks were observed about 6 months after placement. The WFA2 used in this work had a CaO content of 57.9% and was stored in the laboratory for two months before being used in the concrete. Although no detrimental effects were found, the initial swelling of the concrete indicates its reactivity. In contrast, the reactivity of WFA found in this work is beneficial because it reduces the deformations caused by drying shrinkage. This indicates that WFA may have a potential application in concrete as an expansive agent.

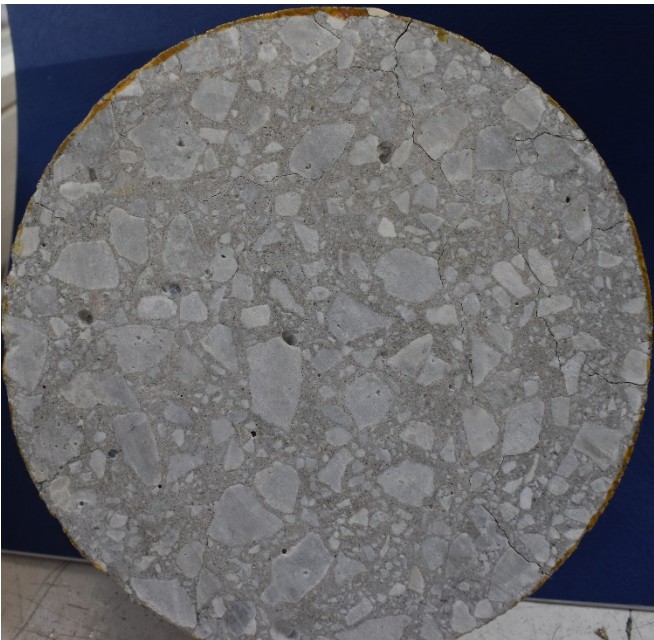

**Figure 8.** Detrimental cracking of a cylindrical concrete specimen with 30% wood fly ash (as a cement replacement) containing 57.6% CaO (results of a previous study [82]).

## 4. Conclusions

This paper presents the results of long-term tests on concrete made with two types and two proportions of WFA as a substitute for part of the cement. The influence of the properties and composition of WFA on the compressive strength, permeability, and shrinkage of the concrete was analyzed over a period of one year. The following conclusions can be drawn from the experiments:

- Depending on the type of WFA, the 15% WFA content reduced the 28-day strength of the concrete by 6 to 14%, while the long-term strength was reduced by 8 to 17%.

- At 30% WFA, the compressive strength at 28 days of age was 25 to 37% lower compared to the reference mix.
- When testing the drying shrinkage of the mixes during more than one year, a significant decrease in shrinkage was observed in the mixes with 30% WFA compared to the reference mix, but in the case of WFA2, the swelling deformation was also pronounced at an early age, which was probably caused by a significant amount of free MgO and free CaO; the TG analysis showed that the mixes with WF2 contained brucite, which can also partially cause the increase in the initial volume.
- WFA1, at a 15% content, unlike WFA2, did not have a large contribution to shrinkage reduction compared to the reference mix.

The potential application of high CaO WFA as a shrinkage reducing admixture requires the determination of the optimum dosage of WFA to minimize the shrinkage and strength decrease. According to the results presented in this work, this could be around 15% of cement replacement. For future applications, it would be useful to know the relationship between storage time and chemical composition of WFA.

**Author Contributions:** Conceptualization, methodology, formal analysis, investigation, I.G., N.Š. and M.S.; Writing—original draft preparation, I.G.; Writing—review and editing, I.G., M.S. and N.Š.; Visualization, I.G.; Project administration, N.Š.; Funding acquisition, N.Š. All authors have read and agreed to the published version of the manuscript.

**Funding:** This research was performed as part of the research project IP-2016-06-7701 'Transformation of Wood Biomass Ash into Resilient Construction Composites', which was funded by the Croatian Science Foundation.

**Institutional Review Board Statement:** Not applicable.

**Informed Consent Statement:** Not applicable.

**Data Availability Statement:** Not applicable.

**Conflicts of Interest:** The authors declare no conflict of interest.

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
