# Peer review of "Long-Term Behavior of Concrete Containing Wood Biomass Fly Ash"

_applsci, doi:10.3390/app122412859_

Round 1

Reviewer 1 Report

The study investigated two types of wood ash as cement replacement at contents of 15% and 30% for concrete production. Fresh and hardened-state proprieties such as slump, density, air content, compressive strength, capillary absorption, gas permeability, and drying shrinkage were assessed. Overall, the manuscript is well structured and has the originality required for publication. However, some specific comments need to be addressed:

1) The introduction is excellent and presents the problem and uniqueness of the work well.

2) “WFA was stored in the laboratory for a period of about two months before being 89 used in concrete.” Remove the sentence that has no importance for the study.

3) “For capillary absorption measurements, the side surface in contact with water was coated with epoxy resin” Insert an image of the specimen.

4) “The calculated initial temperature is given in Table 3”. Update table numbering

5) In discussing the results, the authors need to comment on how the particle size distribution, composition, and LoI results impacted the different behaviors of WFA1 and WFA2.

6) Figure 5 would be more appropriate in the materials and methods section than the results section.

7) “From each prism, 3 concrete cubes with 240 a side length of 10 cm were sawn (Figure 5). Thus, a total of 9 specimens from each mix 241 were tested. At the time of testing, the specimens were 2 years old. The compressive 242 strength measured on cubes with a side length of 10 cm was converted to the strength of 243 a 15 cm cube by multiplying by the coefficient 0.95.” Insert in materials and methods section.

8) “For the TG analysis, concrete powder was extracted from the samples of 2-year-old concrete. To avoid a possible carbonated surface, the powder was extracted with a Profile grinder apparatus (Germann Instruments) from a depth of 10 to 17 mm. Thermogravimetric analysis was performed on the samples of 50±5 mg heated from 30°C to 1000°C at a rate of 10°C/min and a nitrogen flow of 40 mL/min using a TGA 55 instrument (TGA 55, TA Instruments).” Insert in materials and methods section.

9) “A smaller peak in the range between 400-450°C indicates the decomposition of portlandite, while the large peak between 600 and 800°C belongs to the decomposition of dolomite” Insert reference to these ranges.

10) “The compressive strength of concrete with 15% WFA at 2 years of age increases by about 1% compared to the 28-day strength of concrete with the same composition, while at 30% WFA it increases compressive strength by about 3 to 12%.” Remove sentence. An increase of just 1% in compressive strength can be considered statistically equal.

11) Insert a closing comment in the conclusions. Highlight suggestions for future studies (continue to evaluate the possibility of using WFA as an expansive agent).

Reviewer 2 Report

This research is well written and the results support the conclusion of this manuscript. However, it is not ready to be published in this current form because it has a few faults. My comments are as follows:

(1) Abstract. Abstract is not well presented and not clear. Therefore it should be rewrite and contain more highlight the novelty of this study.

(2) Intruduction. The issues of sustainable construction and the use of modern materials to satisfy this postulate should be described more specifically in conjunction with new solutions in this field, in addition to wood fly ash. It shoud be noted that, there are many new pozzolanic materials and possibilities to positively influence the mechanical parameters of cementitious composites. Therefore, it is suggested to discuss several other additives and noadditives and, at least the following new articles from this topic should be discussed and cited:

- Mechanical Performance of Date-Palm-Fiber-Reinforced Concrete Containing Silica Fume, Buildings 2022.

- Nano-Silica-Modified Concrete: A Bibliographic Analysis and Comprehensive Review of Material Properties, Nanomaterials 2022.

- „Nanoparticle-reinforced building materials with applications in civil engineering", Advances in Mechanical Engineering, 2020.

(3) Materials. Besides chemical composition please provide also the phase composition of binders used. Are there any phase differences between WFA1 and WFA2 ?

(4) Results. Please explain quite big variations of test results, presented on Figs. 2 and 3. Whether constructive conclusions can be drawn from such large dispersions ?

(5) Discussion. The phenomenon of resulting cracks in the samples is not well explained in the article, see Fig. 8.

(6) Application. The summary lacks specific practical recommendations resulting from the obtained results.

Round 2

Reviewer 1 Report

Accept in present form

Reviewer 2 Report

I have no further comments.